# Use of an Unmanned Aircraft System to Quantify $NO_x$ Emissions from a Natural Gas Boiler

Brian Gullett[1], Johanna Aurell[2], William Mitchell[1], Jennifer Richardson[3]

[1]US Environmental Protection Agency, Office of Research and Development, Research Triangle Park, North Carolina, 27711, USA; [2]University of Dayton Research Institute, Dayton, Ohio, 45469-7532, USA; [3]The Dow Chemical Company, Midland, Michigan, 48667, USA.

*Correspondence to*: Brian Gullett (gullett.brian@epa.gov)

**Abstract**

Aerial emission sampling of four natural gas boiler stack plumes was conducted using an unmanned aerial system (UAS) equipped with a light-weight sensor/sampling system (the "Kolibri") for measurement of nitrogen oxide (NO), and nitrogen dioxide ($NO_2$), carbon dioxide ($CO_2$), and carbon monoxide (CO). Flights (n = 22) ranged from 11 to 24 minutes duration at two different sites. The UAS was maneuvered into the plumes with the aid of real-time $CO_2$ telemetry to the ground operators and, at one location, a second UAS equipped with an infrared/visible camera. Concentrations were collected and recorded at 1 Hz. The maximum $CO_2$, CO, NO, and $NO_2$ concentrations in the plume measured were 10,000 ppm, 7 ppm, 27 ppm, and 1.5 ppm, respectively. Comparison of the $NO_x$ emissions between the stack continuous emission monitoring systems and the UAS/Kolibri for three boiler sets showed an average of 5.6 % and 3.5 % relative percent difference for the run-weighted and carbon-weighted average emissions, respectively. To our knowledge, this is the first evidence for the accuracy performance of UAS-based emission factors against a source of known strength.

Keywords: Emissions, natural gas, boiler, unmanned aircraft system, drone, continuous emission monitoring

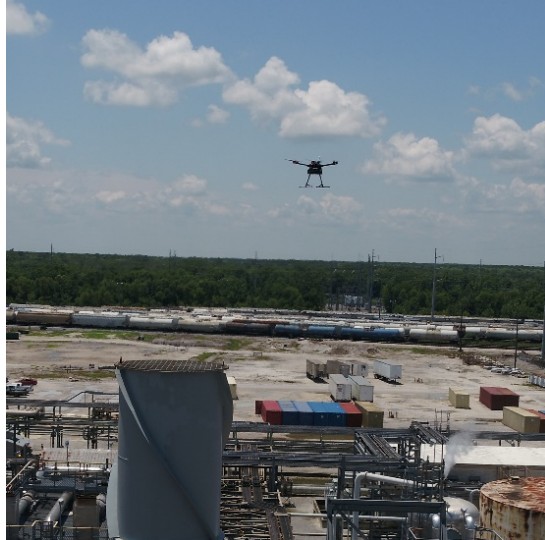

TOC Art

# 1    Introduction

Aerial measurement of plume concentrations is a new field made possible by advances in Unmanned Aircraft Systems (UAS, or "drones"), miniature sensors, computers, and small batteries. The use of a UAS platform for environmental sampling has significant advantages in many scenarios in which access to environmental samples are limited by location or accessibility.  Hazards to equipment and personnel can also be minimized by the mobility of the UAS as well as their ability to be remotely operated away from hazardous sources. UAS-based emission samplers have been used for measurement of area source gases (Neumann et al., 2013; Rosser et al., 2015; Chang et al., 2016; Li et al., 2018), point source gases (Villa et al., 2016), aerosols (Brady et al., 2016), black carbon particles (Craft, 2014), volcanic pollutants  (Mori et al., 2016), particle mass (Peng et al., 2015), and particle number concentrations (Villa et al., 2016).

UAS-based emission measurements are particularly suited for area source measurements of fires and can be used to determine emission factors, or the mass amount of a pollutant per unit of source operation, such as mass of particulate matter (PM) per mass of fuel (e.g., biomass) burned. These values can be converted into emission rates, such as mass of pollutant per unit of energy (e.g., g $NO_x$ $kJ^{-1}$). These determinations typically rely on the carbon balance method in which the target pollutant is co-sampled with the major carbon species present and, with knowledge of the source's fuel (carbon) composition, the pollutant to fuel ratio or an emission rate/factor, can be calculated.

For internal combustion sources that have a process emission stack, downwind plume sampling can use the same method. When combined with the source fuel supply rate and stack flow rates (to determine the dilution rate), measurements comparable to extractive stack sampling may be possible. To our knowledge, determination of emission factors from a stack plume using a UAS-borne sampling system has not previously been demonstrated. The goal of this effort was to compare $NO_x$ measurements obtained by UAS-borne emission samplers with those from concurrent CEM measurements. While not necessarily obviating the need for CEMs for regulatory compliance, the use of UAS-based measurements could provide a safe and fast method of checking emissions that does not require personnel and equipment to access elevated stacks for periodic CEM verification. More importantly, however, the comparison of UAS-based emission measurements against a source of known CEM-determined concentration allows the accuracy of this new type of measurement to be assessed. Demonstrating the efficacy of these measurements would then open their applicability to other less understood sources that are not amenable to conventional CEM sampling, such as open fires, industrial flares, and gas releases.

The feasibility of downwind plume sampling using a sensor-equipped UAS was tested on industrial boilers at the Dow Chemical Company (Dow) facilities in Midland, Michigan (MI) and St. Charles, Louisiana (LA). The sensor system was designed and built by the EPA's Office of Research and Development and the UAS was owned and flown by the Dow Corporate Aviation Group. To determine the comparative accuracy of the measurements, the UAS-based emission factor was compared with the stack continuous emission monitoring systems (CEMS). The target pollutants were nitrogen oxide (NO) and nitrogen dioxide ($NO_2$) to mimic the stack CEMS measurement methods. Carbon as carbon dioxide ($CO_2$) and carbon monoxide (CO) were measured on the UAS for the carbon balance method.

# 2    Materials and Method

Plume sampling tests were conducted on two natural-gas-fired industrial boilers located at Dow's Midland, Michigan and St. Charles, Louisiana facilities.  The Midland boilers are firetube type boilers using low pressure utility supplied natural gas.  They are equipped with low $NO_x$ burners and utilize flue gas recirculation to reduce stack $NO_x$ concentrations. The Midland facility burned natural gas with a higher heating value (HHV) of 9,697 kcal $m^{-3}$ (1089 British Thermal Unit (BTU)/$ft^{-3}$). The two tested stacks are 14 m above ground level and 7 m apart. To

avoid sampling overlapping plumes, only a single boiler was operating during the testing. The St. Charles boilers are
D-type water package boilers using natural gas fuels (high pressure fuel gas (HPFG) and low pressure off-gas
(LPOG)). They are equipped with low $NO_x$ burners with flue gas recirculation to reduce stack $NO_x$ concentrations.
The boiler stacks are about 20 m apart and reach over 20 m in height above ground level. The St. Charles facility
burned natural gas under steady state conditions with a composition of 77.12 % $CH_4$, 2.01 % $C_2H_6$, and 19.91 % $H_2$
and a HHV of 7,845 kcal m$^{-3}$ (881 BTU ft$^{-3}$). Both boilers were operational during aerial sampling, but the wind
direction and UAS proximity to the target stack precluded co-mingling of the plumes.
Air sampling was accomplished with an EPA/ORD-developed sensor/sampler system termed the "Kolibri". The
Kolibri consists of real-time gas sensors and pump samplers to characterize a broad range of gaseous and particle
pollutants. This self-powered system has a transceiver for data transmission and pump control (Xbee S3B, Digi
International, Inc., Minnetonka, MN, USA) from the ground-based operator. For this application, gas concentrations
were measured using electrochemical cells for CO, NO, and $NO_2$ and a non-dispersive infrared (NDIR) cell for $CO_2$
(Table 1). All sensors were selected for their applicability to the anticipated operating conditions of concentration
level and temperature as well as for their ability to rapidly respond to changing plume concentrations due to
turbulence and entrainment of ambient air. Each sensor underwent extensive laboratory testing to verify
performance and suitability prior to selection for the Kolibri. Tests included sensor performance (linearity, drift,
response time, noise, detection limits) in response to anticipated field temperatures, pressure, humidity, and
interferences. Additional information from the manufacturers on sensor performance is available from the links in
Table 1. In anticipation of temperatures as low as 0°C at the Midland site and to avoid daily temperature
fluctuations, insulation was added to the Kolibri frame and the sampled gases were preheated prior to the sensor
with the use of a heating element and micro fan inside the Kolibri. All sensors were calibrated before each sampling
day under local ambient conditions. After sampling was completed, the sensors were similarly tested to assess
potential drift.
Concentration data were stored by the Kolibri using a Teensy USB-based microcontroller board (Teensy 3.2, PJRC,
LLC., Sherwood, OR, USA) with an Arduino-generated data program and SD data card. All four sensors underwent
pre- and post-sampling two- or three-point calibration using gases (Calgasdirect Inc., Huntington Beach, CA, USA)
traceable to National Institute of Standards and Technology (NIST) standards.
**Table 1. UAS/Kolibri Target Analytes and Methods**

| Analyte | Instrument, Manufacturer's Data Link | Frequency | Cal. Gases (ppm) Midland | Cal Gases (ppm) St. Charles |
|---|---|---|---|---|
| $CO_2$ | SenseAir $CO_2$ Engine K30, NDIR[a] https://www.co2meter.com/products/k-30-co2-sensor-module | Continuous, 1 Hz[b] | 408, 990 | 392, 996, 5890 |
| CO | E2v EC4-500-CO, Electrochemical cell https://www.sgxsensortech.com/content/uploads/2014/07/EC4-500-CO1.pdf | Continuous, 1 Hz | 0[c], 9.67, 50.6 | 0, 9.9, 51.8 |
| NO | NO-D4, Electrochemical cell http://www.alphasense.com/WEB1213/wp-content/uploads/2013/10/NOD4.pdf | Continuous, 1 Hz | 0, 2.1, 41.4 | 0, 2.1, 40.4 |
| $NO_2$ | NO2-D4, Electrochemical cell http://www.alphasense.com/WEB1213/wp-content/uploads/2020/09/NO2-D4.pdf | Continuous, 1 Hz | 0, 2.1, 10.4 | 0, 1.9, 10.4 |

[a]Non-dispersive infrared. [b]Hz – hertz. [c]Zero (0) cal gas = air.

The NO sensor (NO-D4) is an electrochemical gas sensor (Alphasense, Essex, UK) which measures concentration by changes in impedance. The sensor has a detection range of 0 to 100 ppm with resolution of < 0.1 RMS noise (ppm equivalent) and linearity within ±1.5 ppm error at full scale. The NO-D4 was tested to have a response time to 95 % of concentration ($T_{95\%}$) of 6.3±0.52 seconds and a noise level of 0.027 ppm. The temperature and relative humidity (RH) operating range is 0 to +50 °C and 15 to 90 % RH, respectively.

The $NO_2$ sensor (NO2-D4) is an electrochemical gas sensor (Alphasense, Essex, UK) which likewise measures by impedance changes. It has a $NO_2$ detection range of 0-10 ppm with resolution of 0.1 RMS noise (ppm equivalent) and linearity error of 0 to 0.6 ppm at full scale. Its $T_{95\%}$ was measured as 32.3±3.8 seconds with a noise level of 0.015 ppm. The temperature and RH operating range is 0 to +50 °C and 15 to 90 % RH, respectively.

Laboratory calibration testing prior to field measurements on both the NO-D4 and NO2-D4 sensors outputs showed their responses to be linearly proportional ($R^2 > 0.99$) over the range of 4- and 5-point calibration gas concentrations. The response times of both sensors were derived using the maximum reference concentration of 47.81 ppm for NO and 10.46 ppm of $NO_2$. The times to reach 95% of the reference concentration, $t_{95}$, were 6.3 and 32.3 sec (RSD (8.2% and 11.8%), respectively, for the NO-D4 and NO2-D4 sensors. These response times are both shorter than those measured simultaneously in the laboratory by a CEM (Ametek 9000$^{RM}$, Pittsburgh, PA, USA) at 37 and 50 sec, respectively, for NO and $NO_2$.

The $CO_2$ sensor ($CO_2$ Engine® K30 Fast Response, SenseAir, Delsbo, Sweden) is an NDIR gas sensor and the voltage output is linear from 400 to 10,000 ppm. The temperature and RH operating range is 0 to +50 °C and 0 to 90 % RH, respectively. The $CO_2$-K30 sensor was measured to have a $t_{95\%}$ response time at 6000 ppm $CO_2$ of 9.0 ± 0.0 seconds and having a noise level of 1.6 ppm. The response time was 4 sec longer than compared to $CO_2$ measured by a portable gas analyzer (LI-820, LI-COR Biosciences, Lincoln, NE, USA). The sensor and the LI-820 showed good agreement as the measurements showed a $R^2$ of 0.99 and a slope of 1.01.

The CO sensor (e2V EC4-500-CO, SGX Sensortech Ltd, High Wycombe, Buckinghamshire UK) is described more fully elsewhere (Aurell et al., 2017; Zhou et al., 2017). In previous sensor evaluation tests with laboratory biomass burns (Zhou et al., 2017) with CO ranging between 0 and 250 ppm, the sensor was compared to simultaneous measurements by a CO CEM (CAI Model 200, California Analytical Instruments Inc., Orange, CA, USA). The concentration measurements had an $R^2 = 0.98$ and a slope of 1.04, indicating the level of agreement between the two devices. The $t_{90}$ was measured as 18 s while comparison of the time-integrated CO concentration differences with the CAI-200, rated at $t_{90} < 1$ s, were only 4.9%.

Variations of the Kolibri sampling system allow for measurement of additional target pollutants. These include particulate matter (PM), polycyclic aromatic hydrocarbons (PAHs), volatile organic compounds (VOCs) including carbonyls, energetics, chlorinated organics, metals from filter analyses, and perchlorate (Aurell et al., 2017; Zhou et al., 2017).

At both facilities the aviation team from Dow flew their DJI Matrice 600 UAS, a six-motor multicopter (hexacopter), into the plumes with EPA/ORD's Kolibri sensor/sampler system attached to the undercarriage (Figure 1). In this configuration of sensors, the Kolibri system weighed 2.4 kg. Typical flight elevations at Midland and St. Charles were 21 and 32 m above ground level (AGL), respectively, and flight durations ranged from 9 to 24 min. At the St. Charles location, the UAS pilot was approximately 100 m from the center point of the two stacks, easily allowing for line of sight operation. A telemetry system on the Kolibri provided real time $CO_2$ concentration and temperature data to the Kolibri operator who in turn advised the pilot on the optimum UAS location.

CEMS on the boiler stacks produced a continuous record of $NO_x$ emission and $O_2$ concentrations.  Stack and CEMS
types located at the Midland and St. Charles facilities are shown in Table 2. The stack $NO_x$ analyzer uses a
chemiluminescence measurement with a photomultiplier tube and is capable of split concentration range operation:
Low (0-180 ppm) and High (0-500 ppm). Its response time is reported as 5 sec. The $O_2$ analyzer uses a zirconium
oxide cell with a measurement range of 0 to 25% and a reported $t_{95}$ of < 10 sec.

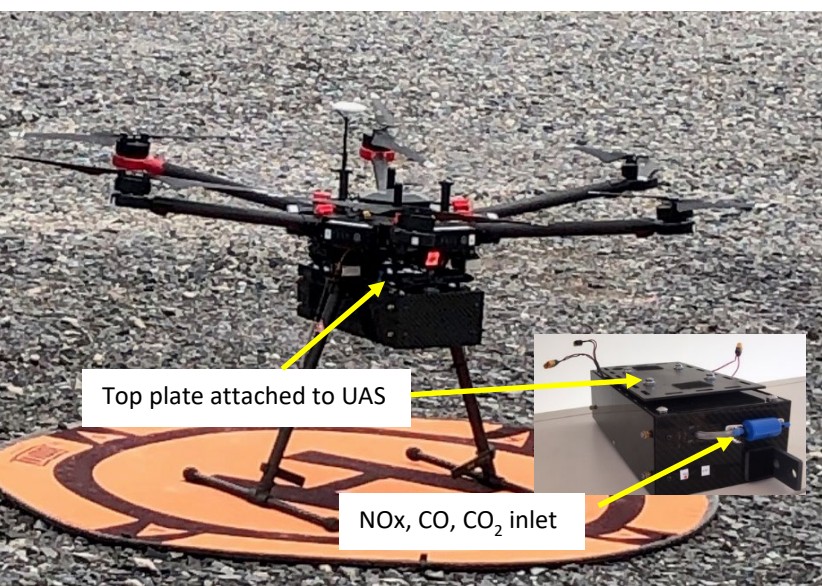

Figure 1. Dow UAS with Kolibri attached to the undercarriage.

**Table 2. CEMS Instruments at both Dow locations.**

| Gas Measured | Midland CEMS | St. Charles CEMS |
|---|---|---|
| $O_2$ | Gaus Model 4705 | ABB/Magnos 106 |
| $NO_x$ | Thermo Model 42i-HL | ABB/Limas 11 |


The plant CEMS undergo annual relative accuracy audit testing (NSPS Subpart Db, Part 70) using US EPA Method
7E (2014) for $NO_x$ and US EPA Method 3A (2017a) for $O_2$. Calculation of $NO_x$ emissions use the appropriate F
factor, a value that relates the required combustion gas volume to fuel energy input, as described in US EPA Method
19 (2017b). Flue gas analysis for $O_2$ and $CO_2$ are performed in accordance with US EPA Method 3A (2017a) using
an infrared analyzer to allow for calculation of the flue gas dry molecular weight.
The CEMS and UAS/Kolibri data were reduced to a common basis for comparison of results. Emission factors, or
mass of $NO_x$ per mass of fuel carbon burned, and emission rates, or mass of $NO_x$ per energy content of the fuel,
were calculated from the sample results. The determination of emission factors, mass of pollutant per mass of fuel
burned, depends upon foreknowledge of the fuel composition, specifically its carbon concentration, and its supply
rate. The carbon in the fuel is presumed for calculation purposes to proceed to either $CO_2$ or CO, with the minor
carbon mass in hydrocarbons and PM ignored for this source type. Concurrent emission measurements of pollutant
mass and carbon mass (as $CO_2$ + CO) can be used to calculate total emissions of the pollutant from the fuel using its
carbon concentration and fuel burn rate.
The UAS/Kolibri emission factors were calculated from the mass ratio of $NO + NO_2$ with the mass of $CO + CO_2$
resulting in a value with units of mg $NO_x$ $kg^{-1}$ C. $CO_2$ concentrations were corrected for upwind background
concentrations. CEMS values of $O_2$ and fuel flowrate were used to calculate stack flowrate using US EPA Method
19 (2017b). This Method requires the fuel higher heating value and an F factor (gas volume per fuel energy content,
e.g., $m^3$ $kcal^{-1}$ ($ft^3$ $BTU^{-1}$)) to complete this calculation. For natural gas, the F factor is 967 $m^3$ $10^{-6}$ kcal (8,710 $ft^3$ $10^{-6}$ BTU) (Table 19-2, US EPA Method 19 (2017b)). The concentration, stack flowrate, and fuel flowrate data allow
determination of $NO_x$ and C emission rates.

## 3    Results and Discussion

The UAS/Kolibri team easily found the stack plumes at both locations using the wind direction and $CO_2$ telemetry
data transmitted to the ground operator.  Use of an infra-red (IR)/visible camera on a second UAS at St. Charles for
some of the flights aided more rapid location of the plume and positioning of the UAS/Kolibri. Gas concentration
fluctuations were rapid and of high magnitude as observed in a representative trace in Figure 2. $CO_2$ concentrations
to 10,000 ppm were observed; the relatively lower average $CO_2$ concentrations reflect the rapid mixing and
entrainment of ambient air causing dilution.

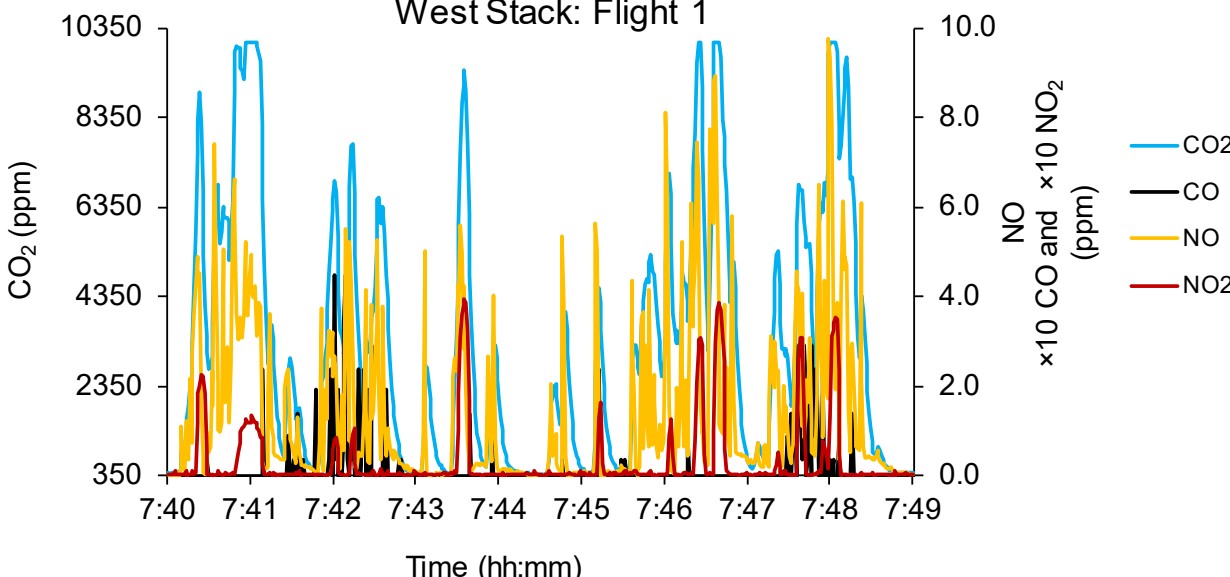


**Figure 2. Example of UAS/Kolibri-measured plume concentrations from the St. Charles West Boiler. Data**
**reported at 1 Hz.**
Sampling data and emission factors from the UAS/Kolibri are shown in Tables 3, 4, and 5 for the Midland, St.
Charles east stack, and St. Charles west stack, respectively.  Eight sampling flights were conducted at the Midland
site, five on the St. Charles East boiler, and nine on the St. Charles West boiler. Both boilers at the Midland site
were operated under the same conditions, so their results have been presented together. Flight times averaged 14 min
(10 % relative standard deviation (RSD)) at the Midland facility and just over 20 min (10 % RSD) at the St. Charles
facility. The shorter flight times in Midland were due to lower UAS battery capacity caused by colder temperatures
(the sampling temperatures in the plume averaged 10±3°C). The average, multi-concentration drift for each of the
sensors, tested at both locations after each sampling day, was less than ±3%. The NO2-D4 sensor showed higher
drift (average 8.6%) at one location for the highest concentration of its calibration gas (10.4 ppm). This had minimal
effect on the emission factor calibrations as the measured $NO_2$ in the plume was actually less than 1 ppm, a range
where the drift was much lower, and $NO_2$ is a minor contributor to the measured $NO_x$ species.
Average plume $NO_x$ concentrations were 0.88±0.32 ppm at Midland and 1.22 ppm and 2.41 ppm at the two St.
Charles boilers with an average RSD of 37 %, 36 %, and 12 %, respectively. The NO emission factor was typically
97 % of the total $NO_x$, with the $NO_2$ providing the minor balance.

**Table 3. Midland UAS/Kolibri Sampling Data and Emission Factors.**

| Date | Flight | Flight time (hh:mm:ss) | | | $NO_2$ | NO | $NO_x$ | Avg. $CO_2$ |
| | # | Up | Down | Total | mg kg$^{-1}$ C | mg kg$^{-1}$ C | mg kg$^{-1}$ C | ppm |
| --- | --- | --- | --- | --- | --- | --- | --- | --- |
| 11/14/2018 | 1 | 10:29:00 | 10:43:00 | 00:14:00 | 201 | 618 | 819 | 1213 |
| 11/14/2018 | 2 | 11:13:04 | 11:28:28 | 00:15:24 | 186 | 624 | 810 | 1138 |
| 11/14/2018 | 3 | 12:54:17 | 13:08:47 | 00:14:30 | 230 | 659 | 889 | 2948 |
| 11/14/2018 | 5 | 13:27:40 | 13:42:05 | 00:14:25 | 99 | 570 | 669 | 4658 |
| 11/15/2018 | 6 | 10:24:20 | 10:39:30 | 00:15:10 | 61 | 394 | 454 | 3703 |
| 11/15/2018 | 7 | 10:41:36 | 10:52:40 | 00:11:04 | 84 | 397 | 481 | 3983 |
| 11/15/2018 | 8 | 10:55:10 | 11:10:10 | 00:15:00 | 126 | 398 | 524 | 4781 |
| **Average** | | | | **00:14:13** | **141** | **523** | **664** | **3203** |
| **Stand. Dev.** | | | | **00:01:28** | **65** | **121** | **179** | **1514** |
| **RSD (%)** | | | | | **10** | **46** | **23** | **27** | **47** |

Flight # 4 excluded from calculations as CO was observed, which originated from a cycling second boiler.

**Table 4. St. Charles East Stack UAS/Kolibri Sampling Data and Emission Factors.**

| Date | Flight | Flight time (hh:mm:ss) | | | $NO_2$ | NO | $NO_x$ | Avg. $CO_2$ |
| | # | Up | Down | Total | mg kg$^{-1}$ C | mg kg$^{-1}$ C | mg kg$^{-1}$ C | ppm |
| --- | --- | --- | --- | --- | --- | --- | --- | --- |
| 07/23/2019 | 1 | 09:49:00 | 10:07:00 | 00:18:00 | 1 | 1442 | 1442 | 2305 |
| 07/23/2019 | 2 | 10:12:00 | 10:34:00 | 00:22:00 | 15 | 1461 | 1476 | 2526 |
| 07/23/2019 | 3 | 10:45:00 | 11:08:00 | 00:23:00 | 5 | 1534 | 1539 | 785 |
| 07/23/2019 | 4 | 11:11:00 | 11:31:00 | 00:20:00 | 101 | 1684 | 1785 | 1082 |
| 07/23/2019 | 5 | 11:52:00 | 12:01:00 | 00:09:00 | 107 | 2110 | 2217 | 1923 |
| **Average** | | | | **00:20:45** | **30** | **1530** | **1560** | **1675** |
| **Stand. Dev.** | | | | **00:02:13** | **47** | **110** | **155** | **869** |
| **RSD (%)** | | | | | **11** | **155** | **7.2** | **9.9** | **52** |

*Flight # 5 was not included in the average as elevated CO concentrations were detected, likely from other sources*
*in the facility.*


| Date | Flight # | Flight time (hh:mm:ss) Up | Down | Total | NO$_2$ mg/kg C | NO mg/kg C | NO$_x$ mg/kg C | Avg. CO$_2$ ppm |
|---|---|---|---|---|---|---|---|---|
| 07/24/2019 | 1 | 07:31:00 | 07:49:00 | 00:18:00 | 25 | 1366 | 1391 | 3221 |
| 07/24/2019 | 2 | 07:52:00 | 08:16:00 | 00:24:00 | 49 | 1263 | 1312 | 3503 |
| 07/24/2019 | 3 | 08:19:00 | 08:38:00 | 00:19:00 | 87 | 1420 | 1507 | 3415 |
| 07/24/2019 | 4 | 09:23:00 | 09:46:00 | 00:23:00 | 65 | 1341 | 1406 | 4509 |
| 07/24/2019 | 5 | 09:49:00 | 10:11:00 | 00:22:00 | 47 | 1296 | 1343 | 4813 |
| 07/24/2019 | 6 | 10:16:00 | 10:36:00 | 00:20:00 | 52 | 1299 | 1351 | 3773 |
| 07/24/2019 | 7 | 10:38:00 | 11:00:00 | 00:22:00 | 53 | 1316 | 1369 | 4194 |
| 07/24/2019 | 8 | 11:51:00 | 12:13:00 | 00:22:00 | 90 | 1460 | 1549 | 3129 |
| 07/24/2019 | 9 | 13:17:00 | 13:39:00 | 00:22:00 | 47 | 1464 | 1511 | 3606 |
| **Average** | | | | **00:21:20** | **57** | **1358** | **1416** | **3796** |
| **Stand. Dev.** | | | | **00:01:56** | **21** | **74** | **86** | **586** |
| **RSD (%)** | | | | **9** | **36** | **5.5** | **6.0** | **15** |


Table 6 presents the average O$_2$ and NO$_x$ measurement results and the fuel supply rate at both locations. Values for
natural gas supply, adjusted for the C$_2$H$_6$ and H$_2$ composition of the St. Charles fuel, were used to calculate the fuel
carbon supply rate. These data allow calculation of the emission factor, mass of NO$_x$ to the mass of carbon, reported
in Table 7.

**Table 5. Multi-Run Average Stack CEMS Data**

| | Midland Both Boilers | St. Charles East Boiler | West Boiler |
|---|---|---|---|
| O$_2$ (%) | 8.2 | 4.9 | 4.5 |
| NO$_x$ (ppm) | 15.7 | 50.4 | 42.9 |
| Fuel rate | 39.3 10$^6$ kJ h$^{-1}$ | 155.2 10$^6$ kJ h$^{-1}$ | 177.8 10$^6$ kJ h$^{-1}$ |


**Table 6. Comparison of Average NOx Emission Factors from CEMS and UAS/Kolibri**

| | Run-Averaged NOx Emission Factor, mg NO$_x$ kg$^{-1}$ C (± 1 std dev) Midland Both Boilers | St. Charles East Boiler | West Boiler |
|---|---|---|---|
| CEMS | 612 ± 10 | 1555 ± 50 | 1303 ± 29 |
| UAS/Kolibri | 664 ± 179 | 1560 ± 155 | 1416 ± 86 |
| RPD: CEM & UAS/Kolibri, % | 8.2 | 0.3 | 8.3 |


The UAS/Kolibri $NO_x$ emission factor for Midland is 8 % higher than the simultaneous CEMS value. For the East
and West boilers at St. Charles, the UAS/Kolibri $NO_x$ emission factor value is <1 % and 8 % higher, respectively,
than the CEMS values. The difference for the UAS/Kolibri in Midland may be attributed in part to the extremely
cold temperature affecting the performance of the electrochemical sensors. The standard deviations for the CEMS
data are based on the run-average $NO_x$ values for each test. These values were calculated based on 10 sec averaging
for the Midland tests, 60 sec averaging in St. Charles, and 1 sec averaging for the UAS/Kolibri. Higher standard
deviations for the UAS/Kolibri are predictable given the rapidly changing values and wide range (~0-10 ppm) of
$NO_x$ data observed in Figure 2. Difference testing for the CEMS and UAS/Kolibri using $\alpha = 0.05$ and assumed
unequal variances indicate that only the West Boiler and UAS/Kolibri are statistically distinct.
The emission rates calculated from the UAS/Kolibri data are 5.6 kg $NO_x \cdot 10^{-3}$ kJ, 14.6 kg $NO_x \cdot 10^{-3}$ kJ, and 13.3 kg
$NO_x \cdot 10^{-3}$ kJ (0.013, 0.034, and 0.031 lbs $NO_x \cdot 10^{-6}$ BTU ), respectively, for the Midland, East St. Charles, and West
St. Charles boilers, below the regulatory standard of 15.5 kg $NO_x \cdot 10^{-3}$ kJ (0.036 lbs $NO_x \cdot 10^{-6}$ BTU). The emission
factors were also calculated as carbon-weighted values to reflect potential differences in plume sampling efficiency
between runs. The Midland, East St. Charles, and West St. Charles UAS/Kolibri emission factors were, respectively,
607, 1525, and 1409 mg $NO_x$ kg$^{-1}$ C. These amounted to relative percent differences of 0.8, 1.9, and 7.8 % between
the CEM and UAS/Kolibri values, for an overall run-weighted average difference of 5.6 %.   The difference between
the CEM readings and those from the Kolibri weighted by the carbon collection amounts, reflecting the success at
being within the higher plume concentrations, was 3.5 %.
**4    Conclusions**
This work reports, to our knowledge, the first known comparison of continuous emission monitoring measurements
made in a stack with downwind plume measurements made using a UAS equipped with emission sensors.
The UAS/Kolibri system was easily able to find and take measurements from the downwind plume of a natural gas
boiler despite lack of any visible plume signature. The telemetry system aboard the Kolibri system reported real time
$CO_2$ concentrations to the operator on the ground, allowing the operator to provide immediate feedback to the UAS
pilot on plume location. Comparison of the CEM data with the UAS/Kolibri data from field measurements at two
locations showed agreement of $NO_x$ emission factors within 5.6 % and 3.5 % for time-weighted and carbon-
collection-weighted measurements, respectively. This work demonstrates the accuracy of a UAS-borne emission
sampling system for quantifying point source strength. These results also have applicability to area source
measurements, such as open fires, which similarly employ the carbon balance method to determine source strength
emission factors.

Data availability. The tabular and figure data are available at the Environmental Dataset Gateway
https://edg.epa.gov/metadata/catalog/main/home.page.

Author contributions. BG was the prime author of the paper and the project lead. JA conducted the Kolibri field
testing and data analysis. WM designed the instrument electronics. JR led the UAS group and field test
arrangements.

Competing interests. The authors declare that they have no conflict of interest.

Disclaimer. The views expressed in this article are those of the authors and do not necessarily represent the views or
policies of the U.S. EPA.

Acknowledgements. Dow's Corporate Aviation Group: Laine Miller, Bryce Young, James Waddell, Jeffrey
Matthews, Chris Simmons, and Anthony DiBiase conducted flights flawlessly. Dow employees Rob Seibert and
Alex Kidd provided technical data and Amy Meskill (Dow), Jennifer DeMelo (Dow), and Dale Greenwell
(EPA/ORD) provided critical logistic support. Patrick Clark (Montrose) reviewed the St. Charles CEMS data.
Financial support. This work was supported through a Cooperative Research and Development Agreement between
the U.S. EPA and The Dow Chemical Company.

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
