# Peer review of "Use of an Unmanned Aircraft System to Quantify NOx Emissions"

_Atmospheric Measurement Techniques, 2020_

## Referee Comment (RC1) · Anonymous Referee #2 · 6 Aug 2020

General Comments

Gullett et al. present stack emission measurements of CO2, CO, NO2, NO with a UAS, using a non-dispersive infrared optical absorption sensor for CO2 and electrochemical sensors for the other gases. They report good agreement of the measured NOx emissions with the continuous emission monitoring system (CEMS) at the test sites. Both, CEMS and the UAV measurement determine the emission via carbon balance calculations.

The authors, however, do not discuss the additional value of their measurement with the UAV over the CEMS measurement. The CEMS measures the same quantity (NOx emissions) with an up to 18 times lower error (see Tab. 7 of the manuscript). The manuscript thereby does not include the description of a scientific goal or application,

like e.g. the validation of the CEMS measurement, studying the dependence of the NOx emission flux on the distance to the emitting stack, or similar. The feasibility of sampling emission plumes with a UAV was shown in the cited studies of e.g. area sources or volcanoes. In many of these works the absence of a CEMS and/or chemical conversion of the measured quantities along the emission plume motivate the studies.

Further, in my opinion, the scope of this journal demands a more detailed description of the 'extensive laboratory testing to verify performance and suitability' (l. 75, 76 of the manuscript) that was carried out. Especially, since electrochemical sensors are known for their rather unstable performance (see e.g. Jiao et al., 2016, https://amt.copernicus.org/articles/9/5281/2016/) and because the sensor calibration site and measurement site in this study have significantly different gas composition (plume and background air). Drifts, dependencies on environmental factors (e.g. temperature, humidity) and cross interferences with other trace gases in amounts that are typical for the sampled emission plume should be documented in the paper.

Without these points thoroughly addressed I advise against publishing the manuscript in AMT.

Specific Comments

1) Faivre-Pierret et al., 1980 measured volcanic gas and particle emissions with an UAV. 2) The influence of the different time constants (response times) of the sensors on the comparability of the individual measurements should be shortly discussed. 3) l.113: The underlying detection techniques of the CEMS measurements should at least be mentioned. 4) A schematic drawing or photograph of the Kolibri setup would be illustrative. 5) The CO and NO2 values in Fig. 2 should be scaled up by at least a factor of 10 in order to be clearly visible. 6) In Fig. 2 it can be observed that the CO2 concentration reaches the upper end of the sensor's range several times. Is this always considered in the average? 7) l.186, 189: If the progressions of the time series as plotted in Fig. 2 were mostly determined by leaving and entering the plume with the

UAV, the standard deviation of this time series would not be a good measure for the measurement error.

References

Jiao, W., Hagler, G., Williams, R., Sharpe, R., Brown, R., Garver, D., Judge, R., Caudill, M., Rickard, J., Davis, M., Weinstock, L., Zimmer-Dauphinee, S., and Buckley, K.: Community Air Sensor Network (CAIRSENSE) project: evaluation of low-cost sensor performance in a suburban environment in the southeastern United States, Atmos. Meas. Tech., 9, 5281–5292, https://doi.org/10.5194/amt-9-5281-2016, 2016.

Faivre-Pierret, R., Martin, D. & Sabroux, J.C., Contribution des Sondes Aérologiques Motorisées à l'Etude de la Physico-Chimie des Panaches Volcaniques, Bull Volcanol (1980) 43: 473. https://doi.org/10.1007/BF02597686

---

## Referee Comment (RC2) · Anonymous Referee #1 · 14 Aug 2020

General Comments: Gullett et al. describe the methods and results for a novel UAS-based sampling approach for stack emissions relative to standard stack continuous emission monitoring system (CEMS). Results indicate good agreement (within 9 percent) for Run-Averaged NOx Emission Factor between UAS and CEMS systems. Error values for UAS-based measurements range from 3 times greater to more than an order of magnitude greater than for CEMS measurement.

The paper would be strengthened by discussion of the implications of differences between methods and the greater error associated with UAS-based measurement. Such a discussion, in turn, may aided by addressing in the Introduction and Conclusions sections, the potential applications of UAS-based measurement for future research or regulatory purposes. The paper could benefit from additional background and discussion of observed sensor performance in the context of known issues relating to sensor performance as affected by atmospheric conditions. The comment from Referee #2 asking for more detail on "extensive testing" is germane, and can be addressed by reference to other publications by the authors (if available) or through the inclusion of descriptions of such testing and data as supplemental materials. Suggestion for reconsideration after "major" revisions is based on author's ability to address above issues.

Specific comments: 1. Suggestion to include closer imager of mounted Kolibri to illustrate location of intake ports 2. Suggestion to include schematic of Kolibri as flown 3. How did the authors treat data values where CO2 readings were at or above the limits of the detectors, and what assumptions were made about error for such readings 4. Vertical axis scale adjustment for CO and NO2

Technical comments: 1. In Table 3, flight 4 is excluded from the table entirely and an explanatory note provided; however, in Table 4, Flight 5 is excluded from calculations (for reasons that appear similar to flight 4's exclusion from the previous table), but its data is retained in the table. Recommendation to leave flight 4 data in table 3 and use common language (e.g. "excluded from calculations") between tables.

---

## Referee Comment (RC3) · Anonymous Referee #1 · 14 Aug 2020

Agreed, but suggest this can be addressed by brief reference to other publications by the authors (if available) or through the use of supplemental material and data (i.e., inclusion in body of paper will distract from focus of research).

---

## Author Comment (AC1) · 10 Oct 2020

RC1 General Comments Gullett et al. present stack emission measurements of CO2, CO, NO2, NO with a UAS, using a non-dispersive infrared optical absorption sensor for CO2 and electrochemical sensors for the other gases. They report good agreement of the measured NOx emissions with the continuous emission monitoring system (CEMS) at the test sites. Both, CEMS and the UAV measurement determine the emission via carbon balance calculations. The authors, however, do not discuss the additional value of their measurement with the UAV over the CEMS measurement. The CEMS measures the same quantity (NOx emissions) with an up to 18 times lower error (see Tab. 7 of the manuscript). The manuscript thereby does not include the description of a scientific goal or application, like e.g. the validation of the CEMS measurement, studying the dependence of the NOx emission flux on the distance to the emitting stack, or similar. The feasibility of sampling emission plumes with a UAV was shown in the cited studies of e.g. area sources or volcanoes. In many of these works the absence of a CEMS and/or chemical conversion of the measured quantities along the emission plume motivate the studies. Further, in my opinion, the scope of this journal demands a more detailed description of the 'extensive laboratory testing to verify performance and suitability' (l. 75, 76 of the manuscript) that was carried out. Especially, since electrochemical sensors are known for their rather unstable performance (see e.g. Jiao et al., 2016, https://amt.copernicus.org/articles/9/5281/2016/) and because the sensor calibration site and measurement site in this study have significantly different gas composition (plume and background air). Drifts, dependencies on environmental factors (e.g. temperature, humidity) and cross interferences with other trace gases in amounts that are typical for the sampled emission plume should be documented in the paper. Without these points thoroughly addressed I advise against publishing the manuscript in AMT.

RESPONSE to general comments: The reviewer states that the manuscript "does not include the description of a scientific goal or application" and we agree, in part, and have modified the manuscript accordingly starting with the final line in the abstract. While the advantages of a UAS platform for emission measurements are mentioned, we now explicitly stated the goal as comparison of UAS-borne emission measurements with concurrent CEM measurements (Introduction, paragraph 3). More explicit statements can lead into policy implications, for which we are unauthorized. While we are aware of previous efforts sampling volcanoes (for example), those measurements are unable to make any comments on the accuracy of the measurements as the volcano source is an unmeasured quantity. In our work we have direct measurements of the source via the in-stack CEMS, allowing us to make definitive comparisons of our ability to determine the source strength with UAS-based measurements. We agree with the reviewer that a detailed description of the sensor performance is important given their inherent potential for instability and inaccuracy. We place our potential sensors through

rigorous in-laboratory testing covering the full range of expected target gas concentrations and then daily pre- and post-measurement calibration checks in the field. These methods are now partially mentioned in the manuscript as well as reference to manufacturers' sensor data sheets – a more extensive description of the testing is outside of the scope of this paper.

Specific Comments

1) Faivre-Pierret et al., 1980 measured volcanic gas and particle emissions with an UAV.

RESPONSE: The current authors have previously measured gases, particles, VOCs, and SVOCs with the use of UAS-born emission equipment: [Aurell et al., 2017 – Atm Env 166]. However, questions about accuracy based upon known source strength and about the potential effects of rotor wash on emission measurements have been incompletely addressed in the literature; this paper answers some of those questions.

2) The influence of the different time constants (response times) of the sensors on the comparability of the individual measurements should be shortly discussed.

RESPONSE: Each sensor has a different response time to changes in concentration; this could be a consideration, depending on how the measurements are used. If comparisons are to be made instantaneously between pollutants, this could have a significant effect, and can be partially mitigated through use of time-averaging of concentrations. Our work took this approach in reporting concentrations, however, this was not critical, as our emission factor calculations were based upon the time-integrated sum of the concentrations over many minutes of sampling. If we had relied upon and reported instantaneous data, the longer response time of the $NO_2$ sensor (32 sec = t95) still would have had minimal effect over instantaneous emission factors as the $NO_2$ concentration was much lower than that of NO and so had little effect upon the emission factor calculation. The NO and $CO_2$ sensors have similar response times (6-9 s) and would have accurately represented the instantaneous emission factor.

3) l.113: The underlying detection techniques of the CEMS measurements should at least be mentioned.

RESPONSE: These are now mentioned in the revised text.

4) A schematic drawing or photograph of the Kolibri setup would be illustrative.

RESPONSE: A close up photo of the Kolibri with text explanations was inserted to Figure 1.

5) The CO and NO2 values in Fig. 2 should be scaled up by at least a factor of 10 in order to be clearly visible.

RESPONSE: We scaled up the NO2 and CO concentrations by 10 times.

6) In Fig. 2 it can be observed that the CO2 concentration reaches the upper end of the sensor's range several times. Is this always considered in the average?

RESPONSE: CO2 peaks exceeded the range of the sensor for 1,108 seconds out of the 16,500 seconds total sampling duration for the 14-run test, or 6.7% of the sampling time. These concentrations, at their range limit, are included in the average emission factor calculations. Exclusion of the NOx and C data during those 1,108 seconds only affects the EFs by 3 %. When the sensors' ranges are exceeded, the UAS pilot is instructed to back off from the source toward a more dilute airstream.

7) l.186, 189: If the progressions of the time series as plotted in Fig. 2 were mostly determined by leaving and entering the plume with the UAV, the standard deviation of this time series would not be a good measure for the measurement error.

RESPONSE: The authors are not clear if the reviewer is referring to the RSD values cited just below Figure 2. If so, we've clarified that the cited RSDs are for flight durations and not for the observed concentrations. In any case, the reviewer is correct regarding his/her observation. Because the plume is mixing and entraining ambient air, there is considerable fluctuation in concentrations even for the stationary UAS, necessitating

time-averaged values. While this is less of an issue for the more homogeneous gas measurements in the stack, the CEMs also use an averaged display value with a rolling average of 60 seconds which further smooths out any fluctuations in concentration.

Referee's References Jiao, W., Hagler, G., Williams, R., Sharpe, R., Brown, R., Garver, D., Judge, R., Caudill, M., Rickard, J., Davis, M., Weinstock, L., Zimmer-Dauphinee, S., and Buckley, K.: Community Air Sensor Network (CAIRSENSE) project: evaluation of low-cost sensor performance in a suburban environment in the southeastern United States, Atmos. Meas. Tech., 9, 5281–5292, https://doi.org/10.5194/amt-9-5281-2016, 2016. Faivre-Pierret, R., Martin, D. & Sabroux, J.C., Contribution des Sondes Aérologiques Motorisées à l'Etude de la Physico-Chimie des Panaches Volcaniques, Bull Volcanol (1980) 43: 473. https://doi.org/10.1007/BF02597686

Authors' References: Aurell, J.; Mitchell, W.; Chirayath, V.; Jonsson, J.; Tabor, D.; Gullett, B., Field determination of multipollutant, open area combustion source emission factors with a hexacopter unmanned aerial vehicle. Atmospheric Environment 2017, 166, 433-440.

RESPONSE The authors recognize that electrochemical sensors do not have the same performance as CEMs. The referees cite Jiao et al. that tested electrochemical sensors under ambient conditions (measurement range of ppb), for a minimum 30-day testing period, and without any calibrations of the sensors. Our manuscript performed calibrations just before the testing and checked the against the same calibration gases just after the test period to establish if the sensors had drifted during the testing period. For clarification we have added sensor calibration information as well as references to more extensive characterization data. The objective of conducting calibration is to ensure that the sensor measures the gas in question within acceptable levels. Calibrations of sensors and CEMs must be performed using certified gases with a known percentage of the gas tested. As such, the calibration gas mixture will always differ from the plume or stack gas mixtures. References to manufactures's technical specification sheets for each sensor have been added; these data sheets include the sensors

performance and cross sensitivity to other gases.

Comment from Referee #1 on Referee #2 comments, RC3

Agreed [with Referee #1], but suggest this can be addressed by brief reference to other publications by the authors (if available) or through the use of supplemental material and data (i.e., inclusion in body of paper will distract from focus of research).

RESPONSE: The authors assume that Referee #1 agrees to Referee #2 comment about "extensive testing" as described in RC2 below. We have added this in the text.
* * *

---

## Author Comment (AC2) · 10 Oct 2020

Referee #1 agreed with Referee #2 that more information regarding sensor quality and characterization was needed in the manuscript. The authors concur and have addressed this in our response to Referee #2.
* * *

---

## Author Comment (AC3) · 10 Oct 2020

RC2 General Comments: Gullett et al. describe the methods and results for a novel UASbased sampling approach for stack emissions relative to standard stack continuous emission monitoring system (CEMS). Results indicate good agreement (within 9 percent) for Run-Averaged NOx Emission Factor between UAS and CEMS systems. Error values for UAS-based measurements range from 3 times greater to more than an order of magnitude greater than for CEMS measurement. The paper would be strengthened by discussion of the implications of differences between methods and the greater error associated with UAS-based measurement. Such a discussion, in turn, may aided by addressing in the Introduction and Conclusions sections, the potential applications of UAS-based measurement for future research or regulatory purposes. The

paper could benefit from additional background and discussion of observed sensor performance in the context of known issues relating to sensor performance as affected by atmospheric conditions. The comment from Referee #2 asking for more detail on "extensive testing" is germane, and can be addressed by reference to other publications by the authors (if available) or through the inclusion of descriptions of such testing and data as supplemental materials. Suggestion for reconsideration after "major" revisions is based on author's ability to address above issues.

RESPONSE to general comments: The authors believe that the standard deviation values don't reflect "error" – our error measures are based on the test runs' emission factor calculations, not the instantaneous variation of the measured concentrations. These run-specific variations in emission factors are quite modest: average of 5.6% (3.5% for carbon-weighted values) for the three boilers. The significant variation in concentrations observed by the sensors within the plume due to mixing of ambient air is compensated for by reliance on whole-run, integrated values of concentrations for determinations of emission factors. Thus, use of UAS for emission determination, much like annual compliance tests with CEMS, would require a series of measurements to arrive at a final value.

The authors are limited to speculate on regulatory prospects due to the policy implications. Additional information on sensor sensitivities is now included in the body.

Specific comments: 1. Suggestion to include closer imager of mounted Kolibri to illustrate location of intake ports

RESPONSE: We have added a labelled photo of the Kolibri.

2. Suggestion to include schematic of Kolibri as flown

RESPONSE: Labelled photo is added.

3. How did the authors treat data values where $CO_2$ readings were at or above the limits of the detectors, and what assumptions were made about error for such readings.

[Figure]

RESPONSE: CO2 peaks exceeded the range of the sensor for 1,108 seconds out of the 16,500 seconds total sampling duration for the St. Charles location (not Midland), or a total of 6.7%. These are included in the average. Exclusion of the NOx and C data during those 1,108 seconds only affects the EFs by 3 %. When the sensors' ranges are exceeded, the UAS pilot is instructed to back off from the source toward a more dilute airstream.

4. Vertical axis scale adjustment for CO and NO2

RESPONSE: We scaled up the NO2 and CO concentrations by 10 times for improved visibility.

Technical comments: 1. In Table 3, flight 4 is excluded from the table entirely and an explanatory note provided; however, in Table 4, Flight 5 is excluded from calculations (for reasons that appear similar to flight 4's exclusion from the previous table), but its data is retained in the table. Recommendation to leave flight 4 data in table 3 and use common language (e.g. "excluded from calculations") between tables.

RESPONSE: We added the data for flight 4 in the revised table 5.

Authors' comments: Data had been erroneously been moved around in Table 5. We have corrected this in the updated table.

---

## Author Response (AR3)

**Use of an Unmanned Aircraft System to Quantify NO$_x$ Emissions from a Natural Gas Boiler**

Brian Gullett[1], Johanna Aurell[2], William Mitchell[1], Jennifer Richardson[3]

**Comments from the editor:**

*The replies address well the laboratory testing approaches, however it still lacks a more detailed description of scientific goals or potential applications of the method, specifically what is the added value of UAV over CEMS? It must be possible to answer this question without running into "unauthorised policy implications"*

**Response**

I have attached the revised PDF of our manuscript, amt-2020-108, "Use of an Unmanned Aircraft System to Quantify NOx Emissions from a Natural Gas Boiler". I have highlighted original sections and used red font to indicate our response to your comments. I understand and appreciate your comments. I have tried to indicate the goals and uses of this technology without implying that the current regulatory policy could or should be supplanted by this technology. It is a bit of a verbal "dance".

B. Gullett

[revised manuscript text omitted]